# Cardiovascular effects of exercise induced dynamic hyperinflation in COPD patients— Dynamically hyperinflated and non- hyperinflated subgroups

**Jozsef Lukacsovits**[1☯], **Gergo Szollosi**[2☯], **Janos T. Varga**[1,3☯] *

**1** Department of Pulmonology, Semmelweis University, Budapest, Hungary, **2** Department of Interventional Epidemiology, Faculty of Public Health, University of Debrecen, Debrecen, Hungary, **3** Department of Pulmonary Rehabilitation, National Koranyi Institute of Pulmonology, Budapest, Hungary

☯ These authors contributed equally to this work.
* varga.janos_tamas@med.semmelweis-univ.hu

**Data Availability Statement:** The raw data is available from: https://figshare.com/articles/dataset/database_xlsx/20101301.

## Abstract

### Introduction

An increase in respiratory rate and expiratory flow limitation can facilitate dynamic hyperinflation (DH), which may cause an element of the intrathoracic pressure in connection with the worsening of venous return, with negative effect on stroke volume (SV) and cardiac output (CO). It has been unclassified, whether poor circulatory adaptation to exercise can be attributed to DH or poor cardio-vascular performance itself in COPD. Only a subset of COPD patients exhibit dynamic hyperinflation during exercise.

### Patients and methods

We designed a study to show how lung mechanical and cardiovascular parameters change in hyperinflated and non-hyperinflated COPD patients during exercise with a new experimental set-up. Thirty-three COPD patients with similar severity of COPD and left ventricular performance (20 men, 13 women, mean±SD age: 65,36±6,95 years) participated. We measured the cardiovascular parameters with a non-invasive device (Finometer-pro) including the left ventricular ejection time index (LVETi) and estimated the change of DH with inspiratory capacity (IC) manoeuvres during exercise.

### Results

Twenty-one subjects exhibited DH (DH group) and 12 did not (non-DH group). The measurement results were given in mean ± SD and difference between the values measured during maximal load and rest also (ΔX = Xmax.load—Xrest). ΔSV and ΔCO were significantly higher in non-DH vs. DH patients (ΔSV: non-DH 9,7 ± 13,22 ml vs. DH -3,6 ± 14,34 ml, p = 0.0142; ΔCO: non-DH 2,26 ± 1,46 l/min vs. DH 0,88 ± 1,35 l/min, p = 0.0024). LVETi was not different between the two groups. Calculated oxygen delivery (DO2) during maximal load was significantly higher in non-DH group.

**Funding:** The authors received no specific funcding for this work.

**Competing interests:** The authors have declared that no competing interests exit.

## Conclusion

We concluded that worse cardiovascular adaptation to exercise of COPD patients can be associated with exercise-induced DH in a similar cardiovascular aged COPD group.

## Introduction

Expiratory airflow limitation can be manifested at rest and during exercise in COPD [1–4]. After reaching a certain speed, flow becomes independent from the increment of pleural pressure, flow limitation appears, which is often manifested in emphysema. Flow limitation can increase and exacerbate the breathing effort and the work of the respiratory muscles, which can generate high intrathoracic pressure. Concomitantly, as ventilation increases during exercise and expiratory time decreases, expiratory airflow limitation makes it impossible to complete a full expiration to the relaxation volume [1]. As a result, end-expiratory lung volume (EELV) can increase, and a phenomenon called dynamic hyperinflation can develop. End-inspiratory volumes approach the total lung capacity, where lower lung and chest wall compliance results higher inspiratory work, which leads to inspiratory muscle fatigue and exercise-limiting dyspnoea in dynamic hyperinflation [1, 4].

O'Donnell et al. have provided confirmatory evidence correlating Borg dyspnea ratings and measurements of inspiratory capacity (IC) to dynamic hyperinflation and a decrease in endurance time during submaximal constant work rate cycle exercise [5].

The circulatory physiological effect of intrathoracic pressure changes has also been the subject of publications [6–9]. High intrathoracic pressure via the compression of the intrathoracic large veins and the right atrium impairs venous return to the right ventricle. The function of the right ventricle differs from that of the left ventricle, because it develops from other cells at fetal age [10], and together with the pulmonary circulation, is located in a variable-pressure compartment in the thorax [7]. The right ventricular performance is significantly dependent on preload by the ventricular filling. Due to the damping effect of the pericardium, the direct effect of smaller intrathoracic pressure fluctuations on the ventricles is not significant. Larger pressure fluctuations mainly affect the left ventricle, while the left ventricle and systemic circulation are located in compartments of different pressures [7].

It is known that the dynamic hyperinflation in COPD has effect on central circulatory limitation. The complex effect of lung mechanics and cardiovascular function is not known on systemic circulation. There was no complex assessment, experimental set-up, which evaluates the lung mechanical, cardiovascular and systemic circulatory effect together. Our hypothesis was to investigate the presence of dynamic hyperinflation and its indirect effect on the circulatory parameters. Questions about the aging of the cardiac muscle were to be answered as well.

## Materials and methods

### Patients

Thirty-three COPD patients participated in a pulmonary rehabilitation program (20 men, 13 women, mean age 65.36±6.95 years, SD. All of the patients were in stable condition, they participated in a 3-week inpatient pulmonary rehabilitation program. Every cardiopulmonary exercise test was performed at the onset of the rehabilitation program. All patients were informed about all part of the study and the patients gave written consent for their participation. Patient's demographics, blood gas analysis and lung function data are presented in Table 1. Exclusion criteria were the presence of any malignant disease, serious congestive heart

**Table 1. Patient demographics, blood gas analysis, and lung function by DH and non-DH groups.**

| Variable | | non-DH | DH |
|---|---|---|---|
| Clinical data | Age (year) | 61.58±7.25 | 67.52±5.91 |
| | Gender | 8 male, 4 female | 12 male, 9 female |
| | Body weight (kg) | 74.67±20.98 | 76.76±20.33 |
| | Body height (cm) | 170.58±9.99 | 165.38±9.54 |
| | BMI | 25.48±5.94 | 28.05±7.06 |
| Blood gas | pH | 7.37±0.03 | 7.38±0.04 |
| | PaO2 (kPa) | 59.23±3.19 | 56.04±9.22 |
| | PaCO2 (kPa) | 43.39±4.82 | 42.62±6.46 |
| | BE (mmol) | -0.18±3.74 | 0.31±2.02 |
| | Hgb (g/dl) | 14.52±1.90 | 14.03±1.25 |
| Lung function | FVC (l) | 2.54±0.65 | 2.07±0.59 |
| | FEV1(l) | 1.21±0.37 | 1.01±0.35 |
| | FVC%pred (%) | 65.20±17.99 | 63.90±11.73 |
| | FEV1%pred (%) | 42.20±14.54 | 39.47±10.96 |
| | FEV1/FVC% | 48.57±13.66 | 48.91±10.24 |

BMI: body mass index, PaO2: partial pressure of the oxygen, PaCO2: partial pressure of the carbon-dioxide, BE: base excess, Hgb: haemoglobin, FVC: forced vital capacity, FEV1: forced expiratory volume in one second.

failure, liver or kidney insufficiency, or any other disease, which can be a contraindication of a cardio-pulmonary exercise test. The study was approved by the Ethical Committee of the National Koranyi Institute of Pulmonology, Budapest, Hungary with registration number of 25/2017. The study was also registered at the ISRCTN registry with ISRCTN13019180 ID. There was no funding, which is related to the clinical study.

## Pulmonary function

According to the Global Lung Function Initiative (GLI) Network, all patients performed spirometry and post-bronchodilator pulmonary function testing (*V*max 229 and Autobox 6200; Sensormedics, Yorba Linda, CA, USA) including spirometry measurements [11]. 400 μg of salbutamol was inhaled 20 minutes before testing.

## Exercise protocol

The patients with COPD performed a continuously increment exercise test (RAMP protocol). We identified the presence of dynamic hyperinflation parameters by inspiratory capacity manoeuvres. The cardiovascular parameters of the patients were measured and recorded continuously during exercise.

Prior to the study, the patients signed an informed consent form, and all patients underwent a physical examination to detect any criteria that might have influenced them for exercise testing. The patients were comfortably seated on the cardio-pulmonary exercise test device, and ECG monitoring, a blood pressure cuff, and a pulse oximeter sensor were used. The face mask in a perfectly fit position to face of the patients was placed [12]. The flow sensor of the respiratory mechanical unit was attached to the face mask and the finger cuff of the finometer was placed on the patient's right middle finger. We started the finometer's measurement before cycling, and it continuously recorded the cardiovascular parameters during the whole test. Lung mechanics were monitored by IC manoeuvres, the process was memorized and checked accurately before the exercise.

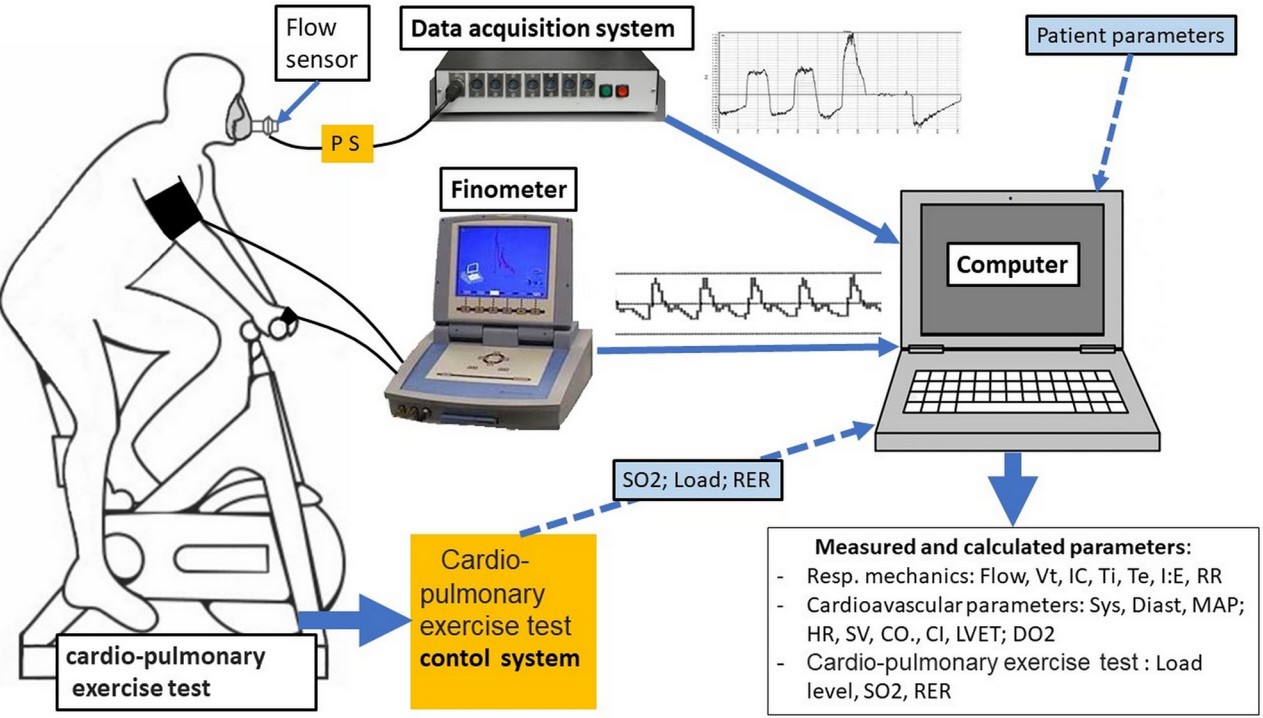

**Fig 1. Experimental set-up.** On-time complex assessment of the exercise physiologic parameters, respiratory flow and systemic circulation. PS: pressure sensor; Flow: respiratory flow; VT: tidal volume; IC: inspiratory capacity; Ti: inspiratory time; Te: expiratory time; I:E: inspiraton/expiration ratio; RR: respiratory rate; Sys: systolic blood pressure; Diast: diastolic blood pressure; MAP: mean arterial pressure; HR: heart rate; SV: stroke volume; CO: cardiac output; CI: cardiac index; LVET: left ventricular ejection time; DO$_2$: oxigen delivery; SO$_2$: oxigen saturation; RER: respiratory exchange ratio.

Lung mechanics data were measured at approximately 1 minute before start of the exercise (to record baseline data), onload phase and 3–5 load levels (depending on the patient's tolerance). The exercise was ended at the maximal exercise tolerance of the patients. Each load level lasted for 2 minutes, and respiratory mechanics data were recorded during the last minute [12].

An IC manoeuvres were performed during each level of load to detect the presence of dynamic hyperinflation. The presence of dynamic hyperinflation was defined by a 5% and at least 100 ml decrease of IC at maximal load compared to rest [12].

The experimental set-up is demonstrated in Fig 1. We detected the airflow, exercise physiologic parameters and cardiovascular parameters simultaneously.

Oxygen saturation was measured in the cardio-pulmonary exercise test with pulse oximeter, and the measured data were downloaded from its memory. No blood gas analysis was performed during the measurements, so only oxygen saturation was used to calculate oxygen delivery (DO2). DO2 is calculated by the following formula: $DO2 = CO \times [(SO2 \times Hb \times 1.39) + (PaO2 \times 0.03)]$. This formula takes into account both haemoglobin-bound and dissolved oxygen, which accounts for only about 1% of DO2. We used a simplified version of the formula that took into account only the oxygen bound by haemoglobin as follows: $DO2 = CO \times (SO2 \times Hb \times 1.39)$.

## Measurement of lung mechanical parameters (description of the respiratory mechanical unit

The respiratory flow of the patients was measured with a flow sensor connected to the face mask of the cardio-pulmonary exercise test. We used a single-use variable orifice flow sensor

(SpiroQuant H; EnviteC USA—New Jersey) [13]. The tubes of the flow sensor were connected to a sensitive differential pressure transducer (HONEYWELL SSCSNBN002NDAA5; sensitivity ±5 cmH2O). The analog electrical signal from the pressure transducer was digitized in a data acquisition device (DAQ; resolution:12 bit; sampling rate: 100 s/s) and then transmitted to a computer, which recorded the data. The airflow values collected in the computer were analysed with a software designed for this purpose (the DAQ and its associated programs were designed and constructed at the Budapest University of Technology and Economics, Faculty of Electrical Engineering and Informatics, Department of Measurement and Information Systems in 2012). Additional parameters were also calculated from the flow curve. The measured and calculated parameters by this device were as follows: respiratory flow (Flow [l/s]), tidal volume (Vt [l]), inspiratory capacity (IC [l]), inspiratory time (Ti [s]), expiratory time (Te [s]), respiratory rate (RR [1/min]) and inspiratory/expiratory ratio (I:E). Prior to each test, the single use flow sensors were checked and calibrated based on the characteristics described by the manufacturer using a Hamilton Galileo ventilator [13].

## Measurement of cardiovascular parameters

Cardiovascular parameters were measured and recorded with a non-invasive device called "Finometer pro" (manufactured by Finapress Medical System BV, The Netherlands in 2002). The "Finometer pro" offers advanced and patented technology [14, 15]. Finapress systems are based on a combination of several technologies: volume clamp with physical technology, brachial artery reconstruction technology, return flow and Modelflow technology. The absolute accuracy of blood pressure measurement was confirmed by comparative studies with intraarterial blood pressure [16–18]. Stroke volume and minute volume were calculated using Modelflow technology based on a three-element Windkessel model [19]. This instrument has been compared in the clinical practice with thermal dilution method [20, 21]. Previous studies have shown that the "Finometer pro" measures the absolute values of stroke volume (SV) and cardiac output (CO) with an accuracy of ± 20% [14, 15], but is able to measure the relative changes in these parameters with an accuracy of ± 8% [14, 15]. This technique can therefore be used to quantify relative changes in minute volume (CO) [14, 15]. The measurement data recorded in the memory of the "Finometer pro" was downloaded with the "Betscope" software and converted to an Excel file. Calibration was performed, before each measurement cycle. The measured and calculated cardiovascular parameters were: systolic arterial pressure (SAP Hgmm), diastolic arterial pressure (DAP Hgmm), mean arterial pressure (MAP Hgmm), heart rate (HR 1 / min), stroke volume (SV ml).), minute volume (CO l / min), left ventricular ejection time (LVET) [20–24].

## Cardio-pulmonary exercise test

Cardio-pulmonary exercise system was used to produce different load levels (Jaeger Vyntus CPX, Wuerzburg, Germany). Planned load levels were determined based on earlier measurement data pertaining to the subjects. From the cardio-pulmonary exercise data, only load levels (L1-L5) and oxygen saturation (SO2[%]) values were used in the present study [24]. Oxygen saturation was measured with the spiroergometer pulse oximeter, and the measured data we have downloaded from its memory.

## Statistical analysis

A pilot study was performed, the results of which were subjected to statistical power analysis to determine the number of samples required. The ClinCalc online program was used for the calculation (https://clincalc.com/stats/samplesize.aspx). Using the percentual change in cardiac

output for the DH and non-DH groups (CO % during max. load: DH group 121,8 ±24,4 SD %; non-DH group: 151,7±25,1 SD %) and taking into account α = 0.05 and 80% statistical power, the calculation showed a total of 22 (11 + 11) minimum number of participants required.

The 33 subjects were divided into two groups according to dynamic hyperinflation during exercise. Twenty-one subjects presented dynamic hyperinflation (Dh group) and 12 did not (non-DH group). Within both groups, we examined the differences between the parameters measured at rest and maximum load, using paired t-test or the corresponding nonparametric Wilcoxon signed-rank test, depending on the normality of sample, which was tested by Shapiro-Wilk test.

Changes in each parameter ($\Delta X = X max.load–X rest$) were calculated and the central values obtained were compared between the two groups (DH and non-DH groups), with two-sample t-test or the according nonparametric, Mann-Whitney U test. With this statistical calculation, we wanted to compare the dynamics of the change on the measured parameters between the DH and non-DH groups, with special regard to the cardiovascular parameters. The Statistical analysis was performed using Stata Statistical Software (version 9.0, Stata Corp, College Station, TX, USA) and $p<0.05$ was considered as significant. All the data are presented as mean ± SD.

The first column of the data files recorded by both the finometer and the respiratory mechanical device was the time of measurement, so it was sufficient to synchronize the internal clock of each instrument before the measurement to synchronize the data.

## Results

All 33 patients completed the study protocol. We designed 5 load levels for each patient, but due to exercise intolerance, approximately half of the patients completed only 3 load levels. Patients were in similar maximum load level in both groups, as evidenced by the fact that no significant differences in RER values were found between the groups (DH: 0.923 ± 0.114 SD; non-DH: 0.935± 0.118; p>0.05).

The mean, standard deviation, and results of the normality test for the absolute values of the respiratory mechanical and cardiovascular parameters in the DH and non-DH groups, and the test results for the differences between rest and maximum load, are shown in Table 2.

Table 3 shows the changes in each parameter (between max load and rest) and the results of their normality analysis. The table also includes the results of a test examining the difference between the two groups. Based on the IC maneuver, dynamic hyperinflation was found in 21 of the 33 patients.

### Respiratory volumes ($V_T$, MV)

In the non-DH group, the mean value of VT was 0.40 l (±0.16) which increased to 0.73 l (±0.30) significantly (p<0.001). In the DH group the mean value was 0.55 l (±0.15) which significantly (p<0.001) increased to 0.72 l (±0.29). The mean difference of VT was 0.34 l (±0.22) in the DH group and 0.18 l (±0.18) in the non-DH group. A significant difference was found (p = 0.030) regarding the deltas.

The MV mean value was 7.77 l (±2.88) which significantly (p<0.001) increased to 21.14 l (±7.21) in the non-DH group. The resting MV was equal to 10.81 l (±3.28) which increased to 24.06 l (±7.09) significantly (p<0.001) in the DH group. The mean difference was equal to 13.37 l (±5.87) in the non-DH group and it was 13.26 l (±5.32) but there was no significant (p = 0.956) difference between the two central values regarding the changes. The changes of the IC compared to resting value was positive in the non-DH and negative in the DH group (Fig 2).

**Table 2. Respiratory volumes, breathing patterns, cardiovascular parameters, oxygen saturation, oxygen delivery and left ventricular ejection time index by DH and non-DH groups.**

| Variable | non-DH n = 12 | | | | | | | DH n = 21 | | | | | | |
|---|---|---|---|---|---|---|---|---|---|---|---|---|---|---|
| | Rest | | | Maximum load | | | Paired p-value | Rest | | | Maximum load | | | Paired p-value |
| | mean | sd | normality | mean | sd | normality | | mean | sd | normality | mean | sd | normality | |
| Vt (l) | 0.40 | 0.16 | 0.86298 | 0.73 | 0.30 | 0.19827 | 0.0002 | 0.55 | 0.15 | 0.43546 | 0.72 | 0.29 | 0.09419 | 0.0003 |
| Ic (l) | 0.82 | 0.31 | 0.01369 | 1.04 | 0.41 | 0.42538 | 0.0015 | 1.33 | 0.52 | 0.17158 | 1.01 | 0.42 | 0.47215 | <0.001 |
| Ti (s) | 1.16 | 0.26 | 0.17473 | 0.78 | 0.22 | 0.90008 | 0.0009 | 1.22 | 0.30 | 0.91735 | 0.69 | 0.21 | 0.32555 | <0.001 |
| Te (s) | 1.89 | 0.49 | 0.76823 | 1.29 | 0.34 | 0.20330 | 0.0017 | 1.83 | 0.42 | 0.43167 | 1.10 | 0.27 | 0.64924 | <0.001 |
| RR 1/min | 20.80 | 5.44 | 0.19612 | 31.13 | 9.68 | 0.01434 | 0.0024 | 20.53 | 4.35 | 0.23663 | 34.74 | 8.41 | 0.18998 | <0.001 |
| I:E | 0.63 | 0.12 | 0.58925 | 0.64 | 0.15 | 0.54419 | 0.7483 | 0.68 | 0.14 | 0.73505 | 0.64 | 0.16 | 0.02020 | 0.2722 |
| Mv l/min | 7.77 | 2.88 | 0.80325 | 21.14 | 7.21 | 0.99626 | <0.001 | 10.81 | 3.28 | 0.14127 | 24.06 | 7.09 | 0.34737 | <0.001 |
| Sys (mmHg) | 138.52 | 28.01 | 0.54024 | 180.82 | 34.18 | 0.93081 | <0.001 | 138.62 | 28.7 | 0.14875 | 155.2 | 43.23 | 0.98390 | 0.0330 |
| Diast (mmHg) | 88.13 | 15.59 | 0.00828 | 104.42 | 20.41 | 0.25117 | 0.0010 | 82.86 | 11.33 | 0.15189 | 94.51 | 21.00 | 0.60397 | 0.0115 |
| MAP (mmHg) | 107.08 | 17.32 | 0.02490 | 135.07 | 24.56 | 0.80125 | 0.0005 | 104.42 | 16.92 | 0.67837 | 120.67 | 29.87 | 0.43500 | 0.0074 |
| HR 1/min | 80.07 | 9.33 | 0.53587 | 107.99 | 16.92 | 0.78620 | 0.0001 | 81.29 | 12.1 | 0.82678 | 103.11 | 14.91 | 0.62201 | <0.001 |
| SV (ml) | 45.92 | 18.04 | 0.11151 | 55.62 | 14.16 | 0.65951 | 0.0274 | 55.05 | 21.21 | 0.07151 | 51.45 | 22.2 | 0.00204 | 0.3737 |
| CO (l/min) | 3.59 | 1.14 | 0.72744 | 5.85 | 1.32 | 0.98761 | 0.0002 | 4.42 | 1.75 | 0.21238 | 5.31 | 2.61 | 0.00090 | 0.0019 |
| CI (l/min/m2) | 1.98 | 0.73 | 0.36512 | 3.25 | 0.98 | 0.46655 | 0.0004 | 2.41 | 0.88 | 0.06203 | 2.88 | 1.25 | 0.00796 | 0.0022 |
| SO2 (%) | 94.58 | 94.58 | 0.00065 | 91.67 | 4.52 | 0.00544 | 0.0020 | 93.19 | 2.52 | 0.27847 | 89.19 | 6.77 | 0.00281 | 0.0008 |
| DO2 (ml) | 766.9 | 214.75 | 0.01344 | 1039.82 | 225.58 | 0.03727 | 0.0098 | 807.5 | 463.35 | 0.00378 | 894.44 | 533.9 | 0.00093 | 0.0361 |

Vt: tidal volume, Ic: inspiratory capacity, Ti: inspiratory time, Te: expiratory time, RR: respiratory rate, I:E: inspiratory to expiratory ratio, Mv: maximal ventilation, Sys: Systolic blood pressure, Diast: Diastolic blood pressure, MAP: mean arterial pressure, HR: hearth rate, SV: stroke volume, CO: cardiac output, CI: cardiac index, SO2: saturation, DO2: oxygen delivery

### Breathing pattern (Ti, Te, RR, I:E)

Regarding the Ti variable, the initial mean value was 1.16 s (±0.26) in the non-DH group which significantly (p<0.001) decreased to 0.78 s (±0.22). The same association was found in the DH group where the parameter was equal to 1.22 s (±0.30) and it decreased to 0.69 l (±0.21) significantly (p<0.001). Mean value of change regarding Ti variable was -0.37 s (±0.29) in the non-DH group which did not differed significantly (p = 0.148) from the mean value observed in the DH group of -0.52 s (±0.27).

In the resting stage the TE was equal to 1.89 s (±0.49) in the non-DH group and it significantly (p = 0.002) decreased to 1.29 s (±0.34) at maximum load level. The same trend was observed in the DH-group, where the value was equal to 1.83 s (±0.42) in the resting stage and it decreased to 1.10 s (±0.27) significantly (p<0.001). The mean difference was -0.6 s (±0.5) in the non-DH group and it was -0.73 s (±0.40) in the DH group. There was no statistically proven difference in the deltas (p = 0.406).

The RR mean value was 20.80/min (±5.44) at the rest stage in the Non-DH group and it increased significantly (p<0.002) to 31.31/min (±9.68) at the maximum load level. The same significant (p<0.001) association was found in the DH group, where the resting stage's median value was 20.53/min (±4.35) and it climbed to 34.74/min (±8.41). The mean difference was greater in the DH group with a mean difference of 14.21/min (±7.95) compared to the non-DH group 10.33/min (±10.61), but the difference was not significant statistically (p = 0.242).

The mean IE was equal to 0.63 (±0.12) in the non-DH group; it was equal to 0.68 (±0.14) in the DH group at the rest stages. In the non-DH group a non-significant (p = 0.748) increase

**Table 3. Differences regarding respiratory volumes, breathing patterns, cardiovascular parameters, oxygen saturation, oxygen delivery and left ventricular ejection time index by DH and non-DH groups.**

| | Delta | | | | | | |
| --- | --- | --- | --- | --- | --- | --- | --- |
| Variable | non-DH n = 12 | | | DH n = 21 | | | non-DH vs. DH p-value |
| | mean | sd | normality | mean | sd | normality | |
| Vt (l) | 0.34 | 0.22 | 0.09438 | 0.18 | 0.18 | 0.67492 | 0.0302 |
| Ic (l) | 0.22 | 0.21 | 0.05346 | -0.32 | 0.17 | 0.17262 | <0.001 |
| Ti (s) | -0.37 | 0.29 | 0.48577 | -0.52 | 0.27 | 0.69255 | 0.1478 |
| Te (s) | -0.6 | 0.5 | 0.99960 | -0.73 | 0.4 | 0.72943 | 0.4064 |
| RR 1/min | 10.33 | 10.61 | 0.15897 | 14.21 | 7.95 | 0.06760 | 0.2418 |
| I:E | 0.02 | 0.16 | 0.37136 | -0.04 | 0.14 | 0.65901 | 0.3388 |
| Mv l/min | 13.37 | 5.87 | 0.80017 | 13.26 | 5.32 | 0.58645 | 0.9558 |
| Sys (mmHg) | 42.3 | 19.77 | 0.67419 | 16.58 | 33.16 | 0.35332 | 0.0206 |
| Diast (mmHg) | 16.29 | 10.05 | 0.38408 | 11.65 | 19.18 | 0.09333 | 0.4435 |
| MAP (mmHg) | 28 | 14.51 | 0.65829 | 16.25 | 24.97 | 0.20864 | 0.1472 |
| HR 1/min | 27.92 | 16.08 | 0.91556 | 21.81 | 11.36 | 0.78459 | 0.2112 |
| SV (ml) | 9.7 | 13.22 | 0.67017 | -3.6 | 14.34 | 0.00072 | 0.0142 |
| CO (l/min) | 2.26 | 1.46 | 0.07508 | 0.88 | 1.35 | 0.02145 | 0.0024 |
| CI (l/min/m2) | 1.27 | 0.89 | 0.09258 | 0.47 | 0.69 | 0.05535 | 0.0071 |
| SO2 (%) | -2.92 | 2.43 | 0.82537 | -4.00 | 5.33 | 0.07740 | 0.5125 |
| DO2 (ml) | 272.92 | 267.49 | 0.25300 | 86.94 | 156.08 | 0.65794 | 0.0248 |

Vt: tidal volume, Ic: inspiratory capacity, Ti: inspiratory time, Te: expiratory time, RR: respiratory rate, I:E: inspiratory to expiratory ratio, Mv: maximal ventilation, Sys: Systolic blood pressure, Diast: Diastolic blood pressure, MAP: mean arterial pressure, HR: hearth rate, SV: stroke volume, CO: cardiac output, CI: cardiac index, SO2: saturation, DO2: oxygen delivery

was seen and a non-significant (p = 0.272) decrease in the DH-group. No significant difference (p = 0.339) was observed regarding the mean changes in the parameter.

## Cardiovascular parameters, oxygen saturation, oxygen delivery and left ventricular ejection time index (Sys, Diast, MAP, SV, CO, CI, SO2, DO2, LVETi)

The mean systolic blood was 138.52 mmHg (±28.01) in the non-DH group at resting stage and it increased to 180.82 mmHg (±34.18) at the maximum load level significantly (p<0.001). The mean value was equal to 138.62 mmHg (± 28,7) in the DH group at resting stage which significantly (p = 0.033) increased to 155.20 mmHg (±43.23) during the exercise till the patients reached maximum load level. The mean difference of systolic blood pressure change was 42.30 mmHg (±19.77) in the non-DH group, and it was 16.58 mmHg (±33.16) in the DH group. A significant (p = 0.021) difference could be observed regarding the two changes compared. The mean diastolic blood pressure was equal to 88.13 mmHg (±15.59) at the rest stage in the non-DH group which increased at maximum load to 104.42 mmHg (±20.41) significantly (p = 0.001). The same significant (p = 0.012) increasing trend was observed in the DH group, where the diastolic blood pressure increased from 82.86 mmHg (±11.33) to 94.51 mmHg (±21.00). No statistically proven difference was seen regarding the two within group changes (p = 0.444), even though the change was greater in the non-DH group (16.26 mmHg (±10.05)) compared to the DH group (11.65 mmHg (±19.18)).

MAP in the non-DH group was 107.08 mmHg (±17.32) in terms of mean value at the rest stage which significantly (p<0.001) increased to 135.07 mmHg (±24.56) at maximum load

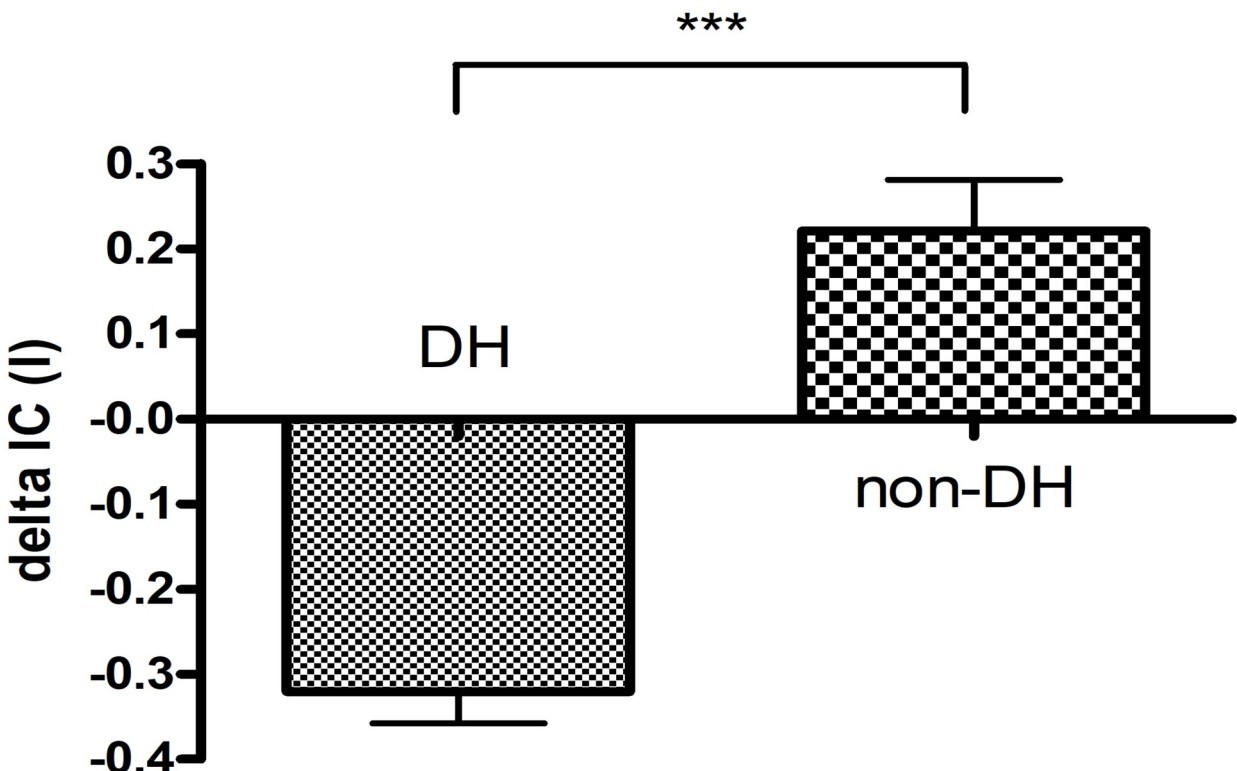

**Fig 2. Comparison of ΔIC value in the DH and non-DH groups.** IC: inspiratory capacity, DH: dynamic hyperinflation; (*** p<0,001).

level. The same trend was observed in the DH group, where it increased from 104.42 mmHg (±16.92) to 120.64 mmHg (±29.87) significantly (p = 0.007). A greater difference was seen in the non-DH group, where the mean value of change was equal to 28.00 mmHg (±14.51) and it was 16.25 mmHg (±24.97) in the DH group, but it was not significant (p = 0.147).

The mean value of SV was 45.92 ml (±18.04) at the resting stage in the non-DH group and it significantly (p = 0.027) increased at maximum load level to a mean value of 55.62 ml (±14.16). In the DH group the mean value was 55.05 ml (±21.21) and it decreased slightly during the exercise to 51.45 ml (±22.20), but this change was not significant (p = 0.374). The mean difference of change in the non-DH group was 9.70 ml (±13.22) and it was -3.6 ml (±14.34) in the DH group, therefore a significant difference was seen regarding the two deltas (p = 0.014).

The cardiac output was 3.59 l/min (±1.14) at the rest stage in the non-DH group which greatly (p<0.001) increased to 5.85 l/min (±1.32) at the end of maximum load. The same increased trend was observed in the DH group, where the resting stage's value was 4.42 l/min (±1.75) and it increased to 5.31 l/min (±2.61) at maximum load in the group of DH, significantly (p = 0.002). The difference in the non-DH group regarding cardiac output was 2.26 l/min (±1.46) in the non-DH group which was statistically higher (p = 0.002) compared to DH group where it was 0.88 l/min (±1.35) (Fig 3).

Mean value of CI was equal to 1.98 l/min/m$^2$ (±0.73) at the rest stage in the non-DH group and it significantly (p<0.001) increased to 3.25 l/min/m$^2$ (±0.98) at maximum load stage; the same trend was seen in the DH group where it increased from 2.41 l/min/m$^2$ (±0.88) to 2.88 l/min/m$^2$ (±1.25). This association was significant (p = 0.002). A significant difference was seen regarding the deltas (p = 0,007), whereas the difference was greater in the non-DH group with

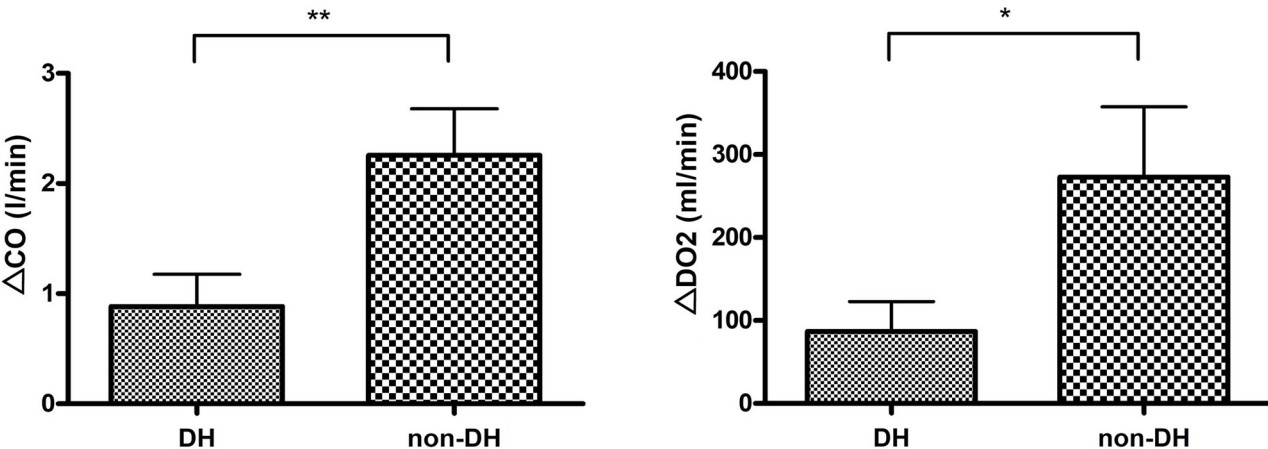

**Fig 3. Comparison of ΔCO and ΔDO₂ value in the DH and non-DH groups.** CO: cardiac output, DO: oxygen delivery, DH: dynamic hyperinflation (** $p<0,01$; * $p<0,05$).

a mean change of 1.27 l/min/m² (±0.89) compared to the change observed in the DH group with a mean value of 0.47 l/min/m² (±0.69).

DO2 was 766.90 ml (±214.75) in the non-DH group and it significantly ($p = 0.010$) increased to 1039.82 ml (±225.58) at the maximum load in terms of mean values. In the DH group, the initial value was 807.50 ml (±463.35) with a significant ($p = 0.036$) increase to 894.44 ml (±533.90) at the end of maximum load level. The mean change in the non-DH group was 272.92 ml (±267.49) and it was 86.94 ml (±156.08) in the DH group, therefore a significant difference was observed between the two values ($p = 0.025$) (Fig 3).

We found that LVETi was in a normal range in all subjects, with no significant difference between the DH and non-DH groups (LVETi; DH group: $422.9 \pm 4.545$ ms SE; non-DH group: $410.3 \pm 5.609$ ms, $p>0.05$).

## Discussion

We conducted a clinical study with thirty-three patients, twenty-one showed a decrease of IC, at least 5% and 100 ml ("DH" group) and twelve had an unchanged or slightly increased IC during exercise ("non-DH" group). The proportion of the DH and non-DH groups correlated with those found in previous studies by Aliverti A et al. and Vogiatzis I et al. [25, 26]. The mean of the maximum loads measured in the two groups showed no significant difference. The respiratory mechanical and cardiovascular parameters were compared in the groups (DH and non-DH) at rest and maximum load level using paired t-test or Wilcoxon test depending on whether the sample met the criteria for normality. We also compared changes of the measured parameters at maximum load relative to the rest value (see Methods) between the DH, and non-DH groups, using t-test or the corresponding non-parametric one. Different physiologic values were detected during exercise between the DH versus non-DH groups. The elevation of the stroke volume, cardiac output and oxygen delivery was lower in the DH group compared to the non-DH group.

Flow limitation during rest and exercise can be manifested in patients with COPD, and consequently static- and dynamic hyperinflation of the chest can develop. A higher level of hyperinflation can result in abnormal lung mechanics with higher end-expiratory (EELV) and end-inspiratory lung volumes (EILV), which have strong connection with reduced exercise

tolerance [27–29]. IC manoeuvres was defined as a marker of the dynamic hyperinflation [30, 31], which we used in our study, also.

A concomitant increase in lung volume results with an increase in peripheral vascular resistance (PVR) can be detected. Overall, positive airway pressure results in a decrease of stroke volume and cardiac output and as consequence a decrease in right ventricular preload and an increase in afterload can develop [6, 7]. Based on the above phenomenon, it can be seen that intrathoracic pressure changes in different respiratory phases have other effects on the right and left ventricles–even at rest–in healthy individuals. In addition to the two main mechanisms discussed above, other minor processes are involved in the interaction between the respiratory and circulatory systems (e.g. neurogenic effects, hormonal effects, ventricular interactions) [6, 7]. The interaction between the respiratory and circulatory systems in various respiratory and cardiovascular pathological conditions is extremely difficult to determine, computer models have been created about this issue, with moderate success [32].

COPD can be associated with cardiac structural and functional abnormalities, lower resting stroke volume can develop. Lower cardiac output and stroke volume in patients with COPD compared to controls was reported at identical level of exercise with the lower absolute oxygen uptake ($\dot{V}o_2$) [33]. We also detected lower cardiac output, stroke volume and oxygen uptake in the COPD group.

When dynamic hyperinflation occurs, intrathoracic pressure becomes higher during exhalation, thereby reducing the right ventricular (RV) preload and caused by increased lung volume PVR increases, leading to an increment in RV afterload. Several studies have been conducted recently to clarify this issue [34, 35].

Smith et al. compared 13 COPD patients with 10 healthy controls during exercise. The load level was adjusted, so that patients consumed the same amount of oxygen. The study found that the COPD group responded to exercise with a significantly lower increase in stroke volume and cardiac output. Another important finding of the study was that the intrathoracic pressure during exercise in COPD group showed significantly greater swing than healthy subjects [34].

Laveneziana P. et al discussed the central hemodynamic effects of dynamic hyperinflation in a recent review. They highlighted that in addition to the central circulatory effects of dynamic hyperinflation, the role of concomitant intrinsic myocardial disease and myocardium aging is unknown [36]. We also focused the myocardial ageing measurement.

Aliverti et al. measured the changes in lung volumes during exercise in COPD patients, using an optoelectronic plethysmograph (OEP). They found that only a subset of COPD patients (of the same severity) presents DH. They defined hyperinflator and euvolemic groups [25]. We have found also only a subset of the COPD patients, who have DH.

Vogiatzis I et al. also studied with OEP the changes in lung volumes during exercise in COPD patients. Based on their results, "early" and "late" dynamic hyperinflator groups were identified. In the "late" hyperinflator group, the development of dynamic hyperinflation started after reaching 66% of the maximum load [26, 37].

In a recent study, Vitacca M. et al concluded that positive end-expiratory pressure (PEEPi, dyn) was weakly correlated to noninvasive measures of lung and respiratory muscle function [38]. To measure dynamic hyperinflation, we used a non-invasive method, which is suitable for identifying dynamic hyperinflation, but less suitable for its quantitative measurement. For this reason, no calculations were performed with the exact value of the IC change, only the presence or absence of dynamic hyperinflation was defined by it.

Ejection distance (ED) is the time interval between the foot of the pressure wave and the beginning of the incisura caused by aortic valve closure [39]. We need to underline that using the ED and the heart rate (HR), we can calculate the LVETi, which is a good indicator of left

ventricular performance [23, 40]. We found that LVETi was in good range in both the DH and non-DH groups, with no significant difference between the DH and non-DH groups, indicating that subjects had good left ventricular function without a significant difference between the two groups. Thus, based on this result, we hypothesized that exercise-induced cardiovascular parameter changes were not significantly affected by left ventricular performance in the studied subjects. We found no significant differences between the two groups in the absolute values of circulatory parameters, neither at rest and exercise. The relative increase in Sys, SV, CO, CI, and DO2 also showed a significantly greater increase in the non-DH group. As the cardiac status of the subjects was similar (as described above), the differences found in SV, CO, and DO2 can be attributed to the central circulatory effect of dynamic hyperinflation on exercise. All of these observations may be adapted for pulmonary rehabilitation programs [41–43].

## Limitations of our study

We do not know the initial resting total lung capacity of the patients, however, we focused not on the resting, but the dynamic change of chest hyperinflation by IC manoeuvres. In the absence of preliminary cardiac ultrasound examination, we do not know if there were any patients with any degree of secondary pulmonary hypertension (PH). The presence of severe PH was unlikely due to the absence of ECG abnormalities and the absence of symptoms and physical abnormalities suggestive of PH.

## Summary

We developed a new alternative experimental set-up to identify lung mechanics and systemic circulatory response together at isotime condition in COPD patients. We could observe the dynamically hyperinflated lung with poor cardiovascular performance in maintained left ventricular function during exercise. A lower level of rise was detected in systemic hemodynamic parameters in the group of COPD patients with dynamic hyperinflation during exercise, although there was no significant difference in left ventricular performance between the two groups. This new type of perspective focuses not only on the consequences of lung mechanics, but also on the impaired systemic circulation as a result of dynamic hyperinflation in patients with COPD. It may have consequence in the generable management of patients with COPD in the future.

## Conclusion

As a clinical implication of the study we developed a new non-invasive and easy to use set-up, which can assess the dynamic hyperinflation, the cardiovascular function and the change in the systemic circulation together. We can use this set-up to the differential diagnosis of dyspnoea in a complex, non-invasive way. We can evaluate the effectiveness of different pharmacotherapeutical and physical treatment intervention in a complex way including cardiopulmonary, circulatory function.

## Future plan

In the future we are planning to use different, sophisticated images to interpret the complex physiological, pathophysiological response, which is going on exercise in a patients with COPD, reduced heart function and heart failure. We would like to more about the anatomical and functional part of the heart in this population.

## Author Contributions

**Conceptualization:** Jozsef Lukacsovits, Gergo Szollosi, Janos T. Varga.

**Data curation:** Gergo Szollosi, Janos T. Varga.

**Formal analysis:** Jozsef Lukacsovits, Gergo Szollosi, Janos T. Varga.

**Investigation:** Jozsef Lukacsovits, Janos T. Varga.

**Methodology:** Jozsef Lukacsovits, Gergo Szollosi, Janos T. Varga.

**Project administration:** Gergo Szollosi.

**Resources:** Janos T. Varga.

**Software:** Jozsef Lukacsovits.

**Supervision:** Janos T. Varga.

**Validation:** Jozsef Lukacsovits, Janos T. Varga.

**Visualization:** Janos T. Varga.

**Writing – original draft:** Jozsef Lukacsovits, Gergo Szollosi, Janos T. Varga.

**Writing – review & editing:** Jozsef Lukacsovits, Gergo Szollosi, Janos T. Varga.

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
