## [Decision Letter · Decision Letter 0]

18 Apr 2022

PONE-D-22-02168Cardiovascular effects of exercice induced dynamic hyperinflation in COPD patients – dynamically hyperinflated and non-hyperinflated  subgroups.PLOS ONE

Dear Dr. Varga,

Thank you for submitting your manuscript to PLOS ONE. After careful consideration, we feel that it has merit but does not fully meet PLOS ONE’s publication criteria as it currently stands. Therefore, we invite you to submit a revised version of the manuscript that addresses the points raised during the review process.

The authors presented a longitudinal study that investigates how lung mechanical and cardiovascular parameters change in hyperinflated and nonhyperinflated COPD patients during physical exercise. Some significant results are presented related to the impact of dynamic exercise hyperinflation on cardiovascular system. I think this results are important in the field and want to encourage the authors to improve their manuscript according to the reviewers' comments. Please, resolve all the reviewer's considerations and suggestions.

We look forward to receiving your revised manuscript.

Kind regards,

Nejka Potocnik

Academic Editor

PLOS ONE

Journal Requirements:

Reviewers' comments:

Reviewer's Responses to Questions

**Comments to the Author**

1. Is the manuscript technically sound, and do the data support the conclusions?

Reviewer #1: No

Reviewer #2: Yes

2. Has the statistical analysis been performed appropriately and rigorously? 

Reviewer #1: No

Reviewer #2: Yes

3. Have the authors made all data underlying the findings in their manuscript fully available?

Reviewer #1: No

Reviewer #2: No

4. Is the manuscript presented in an intelligible fashion and written in standard English?

Reviewer #1: No

Reviewer #2: Yes

5. Review Comments to the Author

Reviewer #1: PONE-D-22-02168: statistical review

SUMMARY. This is a longitudinal study that investigates how lung mechanical and cardiovascular parameters change in hyperinflated and nonhyperinflated COPD patients during physical exercise. The statistical analysis is essentially based on repeated measures anova and Friedman tests for the analysis of cardiovascular parameters at different load levels. I have several concerns about this paper, (see major issues below). I also append below some specific points that should be addressed.

MAJOR ISSUES

1. As said in the abstract, the goal of this paper is to investigate “how lung mechanical and cardiovascular parameters change in hyperinflated and nonhyperinflated patients”. However ANOVA and Friedmann tests do not give a full answer to this question: they just test whether differences between different load levels are significant or not. A fully specified mixed model with random slopes would provide a more comprehensive analysis.

2. Almost nothing is said about the assumptions made. Results of the Kolmogorov-Smirnov test are not displayed and we don’t know which variables have been analyzed by ANOVA and which ones have been analyzed by a Friedman test. In addition, nothing is said about spericity of the ANOVA methods. This lack of material makes the results not reproducible and difficult to compare.

3. Hyperinflation is treated as a factor on which the statistical analysis is conditioned. However, the presence of dynamic hyperinflation is a response observed during the physical exercise. As a result, the whole statistical analysis doesn’t seem quite appropriate. Under this setting, a more natural strategy would be a battery of mixed models where both hyperinflation and cardiovascular parameters are treated as responses.

4. The available covariates described in Table 1 are ignored. The two samples of hyperinflated and nonhyperinflated COPD patients are treated as two homogeneous samples. I’d welcome a comparison between the two samples: significant associations between some covariates and the occurrence of hyperinflation could invalidate the whole analysis.

5. Results section: “the data set was reduced to 3 load levels in each case by averaging the parameters measured at medium load levels.” This is an unnecessary waste of information: repeated measure methods allow time series of different lengths

SPECIFIC POINTS

1. Although the authors state that the data are available without restrictions, data are not attached as supplementary material. Data should be attached to the submission or made availale in a public repository and metadata provided.

2. There are still English typos across the paper: please correct them.

3. Numbering pages and lines would facilitate the reviewer’s job …

Reviewer #2: The manuscript presents some significant results related to the impact of dynamic hyperinflation on cardiovascular system, a research direction that was recently emphasized by Smith et al. (ref [35] in the manuscript), who showed that COPD patients have lower stroke volume and cardiac output during exercise than controls and that hyperinflation has a major impact on central hemodynamics. Comparing two groups of COPD patients with and without dynamic hyperinflation and using noninvasive methods for measuring cardiovascular parameters, the authors showed that even if the cardiac output, stroke volume and oxygen delivery changes had a lower increase at exercise in the hyperinflated group, the left ventricular function (evaluated by the left ventricular ejection time index LVETi) was in good ranges in the two groups, with no significant differences between the groups; the authors are concluding that these results reflect the central circulatory effect of dynamic hyperinflation at exercise.

The intervention is a mixed combination between standard techniques and an original protocol.

The size of the study group was well established.

Acceptance by Ethical Committee is mentioned.

The methods are generally well described; nevertheless there are important aspects of the methods which are detailed elsewhere in the manuscript (method of defining hyperinflation and to calculate LVETi) and should be better mentioned in the “Methods” section:

- Page 9 paragraph 2 “Discussion” [...] decrease of IC, at least 5% and 100 ml (“DH group”) [...]

- Page 11 paragraph 5 [...] Ejection distance (ED) is the time [...] using ED and the heart rate we calculate LVETi [...] - ED is used instead of LVET which was previously mentioned as one of the parameters being measured.

The data underlying the findings in the manuscript were not fully available neither as part of the manuscript nor deposited in a public repository.

I recommend to include in the Results section a table which describes separately the subjects in the 2 groups: DH and non-DH, similar to table 1. It should also include the mean absolute values of cardiac parameters (SV, CO, CI, DO2, LVET and LVETi) at rest and after exercise for the DH and non-DH groups.

The results support in general the conclusion. However, while the Discussion section presents some previously published results, there is little comparison between these publications and the authors' results.

The authors describe also some limitations of their research.

The manuscript needs a more concise organization and a thorough revision for English and grammar. I strongly recommend language editing, as several misspelling are encountered throughout the paper.

Examples:

1. The second paragraph of the introduction does not contain references, and is repeated entirely in the Discussion section (third paragraph).

2. The last sentence in the conclusion should be rewritten:

[…] We can evaluate the effectiveness of different intervention in a complex way including cardiopulmonary function, which including pharmacotherapeutical and physical treatment, also.

3. Lack of citations:

In “Measurement of cardiovascular parameters”

[…] Previous studies have shown that the “Finometer Pro” measures the absolute values […]. This technique can therefore be used to quantify […]

4. There is no uniform style of citations:

Laveneziana et al. instead of Laveneziana P. et al […]

5. There is no uniform use of notation for gases:

DO2 versus DO2

6. The authors are using abbreviations not explained in the text:

PVR page 10 paragraph 3

PEEPi,Dyn page 11 paragraph 4

7. There are no uniform notation for parameters at rest:

Xspont versus Xrest

8. There are repeated paragraphs:

[…] To compare parameters for different load levels, repeated measures ANOVA or Friedman tests were performed. […]

Next paragraph:

[…] Within both groups, we examined the differences between the parameters at different load levels with repeated measures ANOVA or Friedman tests.

6. PLOS authors have the option to publish the peer review history of their article (what does this mean?). If published, this will include your full peer review and any attached files.

Reviewer #1: No

Reviewer #2: No

---

## [Author Response · Author response to Decision Letter 0]

3 Jul 2022

Thank you very much for Your and the Reviewer`s hard work with our manuscript. We incorporated the suggestions and corrected the mistakes in the manuscript, which improved the scientific value of our manuscript. Please find the point-point answers for the reviewer`s questions.

Thank you very much for your work in advance. 

With kind regards,

Janos Varga

Answers for the Editor:

Thank you very much for the information, we corrected the manuscript to the required format.

Thank you very much for your important comment, we added information about in the Methods section.

3.We note that you have indicated that data from this study are available upon request. PLOS only allows data to be available upon request if there are legal or ethical restrictions on sharing data publicly. For information on unacceptable data access restrictions, please see http://journals.plos.org/plosone/s/data-availability#loc-unacceptable-data-access-restrictions.

The study was conducted according to the guidelines of the Declaration of Helsinki and it was approved by the Ethical Committee of the National Koranyi Institute of Pulmonology, Budapest, Hungary with registration number of 25/2017. The study was also registered at the ISRCTN registry with ISRCTN13019180 ID. There was no funding, which is related to the clinical study. Protection of natural persons with regard to the processing of personal data and on the free movement of such data, and re-pealing Directive 95/46/EC (General Data Protection Regulation). All of these data are available in our database network, which can be assessed by the internet.

Review Comments to the Author

Answers for the questions of the reviewer 1:

Thank you very much for work and time in relation with our manuscript.

Please find our answers for your questions.

Reviewer #1: PONE-D-22-02168: statistical review

1. As said in the abstract, the goal of this paper is to investigate “how lung mechanical and cardiovascular parameters change in hyperinflated and nonhyperinflated patients”. However ANOVA and Friedmann tests do not give a full answer to this question: they just test whether differences between different load levels are significant or not. A fully specified mixed model with random slopes would provide a more comprehensive analysis.

Thank you very much for your comment. We propose to address this issue by changing the performed statistical methods, therefore we rephrased the interpretation of the results as well in the manuscript. This is true that ANOVA and Friedmann tests were not designed to answer more comprehensive issues as suggested by the Reviewer, that is why we modified our study design which involved the modification of the time variable into a simpler one for a better comprehensibility. That is why we performed two-sample t-tests and paired t-test for the comparisons if the application criteria of these tests were met. If not, then we performed Mann-Whitney U test and Wilcoxon signed-rank tests. 

Due to properly understand and interpret the physiological effects regarding lung mechanical and cardiovascular parameters among COPD patients during exercises; we excluded the intermediate stages and mainly focused on the differences observed regarding to rest and maximum loaded stages. Since our study did not primarily focus on the comparison of different stages but on the differences observed regarding to the resting and maximum load stages, nonetheless, we performed the multivariate mixed model with two explanatory categorical variables (hyperinflated/non-hyperinflated; load stages) and random slopes (based on the unique identifier) for a deeper analysis; and we found that the time-factor contributes greatly to the observed changes within the groups, which is unambiguous, that is why we ought to change the statistics into a more simple, but applicable to understand the differences observed. We believe that the new study design and the performed statistics will contribute to readers’ contentment. We interpreted the results by using mean and standard deviation a better comprehensibility. 

2. Almost nothing is said about the assumptions made. Results of the Kolmogorov-Smirnov test are not displayed and we don’t know which variables have been analysed by ANOVA and which ones have been analysed by a Friedman test. In addition, nothing is said about spericity of the ANOVA methods. This lack of material makes the results not reproducible and difficult to compare.

Thank you very much for your comment. We modified the tables which now contain the p-values regarding the normality tests. The appropriate parametric or non-parametric statistical tests were chosen based on the application criteria of the statistical tests.

3. Hyperinflation is treated as a factor on which the statistical analysis is conditioned. However, the presence of dynamic hyperinflation is a response observed during the physical exercise. As a result, the whole statistical analysis doesn’t seem quite appropriate. Under this setting, a more natural strategy would be a battery of mixed models where both hyperinflation and cardiovascular parameters are treated as responses.

Thank you very much for your comment. Please see our response to the first comment.

4. The available covariates described in Table 1 are ignored. The two samples of hyperinflated and nonhyperinflated COPD patients are treated as two homogeneous samples. I’d welcome a comparison between the two samples: significant associations between some covariates and the occurrence of hyperinflation could invalidate the whole analysis.

Thank you very much for your comment. We added the new tables for a better comprehensibility of the results.

5. Results section: “the data set was reduced to 3 load levels in each case by averaging the parameters measured at medium load levels.” This is an unnecessary waste of information: repeated measure methods allow time series of different lengths

We resolved this issue in the Results section. We rephrased the affected sentences for better consistency.

SPECIFIC POINTS

1. Although the authors state that the data are available without restrictions, data are not attached as supplementary material. Data should be attached to the submission or made availale in a public repository and metadata provided.

Thank you very much for your comments. The study was conducted according to the guidelines of the Declaration of Helsinki and it was approved by the Ethical Committee of the National Koranyi Institute of Pulmonology, Budapest, Hungary with registration number of 25/2017. The study was also registered at the ISRCTN registry with ISRCTN13019180 ID. There was no funding, which is related to the clinical study. Protection of natural persons with regard to the processing of personal data and on the free movement of such data, and re-pealing Directive 95/46/EC (General Data Protection Regulation). Data Availability All of these data are available in our database network, which can be assessed by the internet.

2. There are still English typos across the paper: please correct them. Thank you very much for your comment. 

Thank you for your comment, we corrected the grammar and the English typos.

3. Numbering pages and lines would facilitate the reviewer’s job …

Thank you for your comment, we added the numbers of the lines and pages to the manuscript.

Answers for reviewer`s 2 questions

Thank you very much for work and time in relation with our manuscript.

Please find our answers for your questions.

1. - Page 9 paragraph 2 “Discussion” [...] decrease of IC, at least 5% and 100 ml (“DH group”) [...]

Thank you for your comment, we added this information to the methods section and the discussion, also.

2. - Page 11 paragraph 5 [...] Ejection distance (ED) is the time [...] using ED and the heart rate we calculate LVETi [...] - ED is used instead of LVET which was previously mentioned as one of the parameters being measured.

Thank you for your comment. LVET=left ventricular ejection time and index is showing us the same parameter. We used the parameter in the methods, result and discussion sections.

3. The data underlying the findings in the manuscript were not fully available neither as part of the manuscript nor deposited in a public repository.

All of these data are available in our database network, which can be assessed by the internet.

4. I recommend to include in the Results section a table which describes separately the subjects in the 2 groups: DH and non-DH, similar to table 1. It should also include the mean absolute values of cardiac parameters (SV, CO, CI, DO2, LVET and LVETi) at rest and after exercise for the DH and non-DH groups.

Thank you for your comment. We added a separate part in the result section and a table for cardiac parameters (SV, CO, CI, DO2, LVET and LVETi).

5. The results support in general the conclusion. However, while the Discussion section presents some previously published results, there is little comparison between these publications and the authors' results.

Thank your for your comment. We added more information about our study compared to previous results.

6. The authors describe also some limitations of their research.

Thank you for your comment, we described a limitation of our study in a correct way.

7. The manuscript needs a more concise organization and a thorough revision for English and grammar. I strongly recommend language editing, as several misspelling are encountered throughout the paper.

Thank you for your comment. We reorganized some part of the manuscript and use a language editing to correct the grammar mistakes.

8. The second paragraph of the introduction does not contain references, and is repeated entirely in the Discussion section (third paragraph).

Thank you for your comment, we corrected this mistake and add reference.

9. The last sentence in the conclusion should be rewritten:

[…] We can evaluate the effectiveness of different intervention in a complex way including cardiopulmonary function, which including pharmacotherapeutical and physical treatment, also.

 Thank you for your comment, we corrected the sentence.

10. Lack of citations:

In “Measurement of cardiovascular parameters”

[…] Previous studies have shown that the “Finometer Pro” measures the absolute values […]. This technique can therefore be used to quantify […]

Thank you for your comment, we corrected it, added references. 

11. There is no uniform style of citations:

Laveneziana et al. instead of Laveneziana P. et al […]

Thank you for your comment, we corrected the style of citations.

12. There is no uniform use of notation for gases:

DO2 versus DO2

Thank you for your comment, we corrected it.

13. The authors are using abbreviations not explained in the text:

PVR page 10 paragraph 3

PEEPi,Dyn page 11 paragraph 4

Thank you for your comment, we corrected the mistake, added explanation of the abbreviation.

14. There are no uniform notation for parameters at rest:

Xspont versus Xrest

Thank you for your comment, we corrected it.

15. There are repeated paragraphs:

[…] To compare parameters for different load levels, repeated measures ANOVA or Friedman tests were performed. […]

Next paragraph:

[…] Within both groups, we examined the differences between the parameters at different load levels with repeated measures ANOVA or Friedman tests.

Thank you very much for your comment, we corrected the statistical methods and the mistake.

---

## [Decision Letter · Decision Letter 1]

31 Aug 2022

Cardiovascular effects of exercise induced dynamic hyperinflation in COPD patients – dynamically hyperinflated and non-hyperinflated subgroups

PONE-D-22-02168R1

Dear Dr. Varga,

We’re pleased to inform you that your manuscript has been judged scientifically suitable for publication and will be formally accepted for publication once it meets all outstanding technical requirements.

Kind regards,

Nejka Potocnik

Academic Editor

PLOS ONE

Additional Editor Comments (optional):

Reviewers' comments:

Reviewer's Responses to Questions

**Comments to the Author**

1. If the authors have adequately addressed your comments raised in a previous round of review and you feel that this manuscript is now acceptable for publication, you may indicate that here to bypass the “Comments to the Author” section, enter your conflict of interest statement in the “Confidential to Editor” section, and submit your "Accept" recommendation.

Reviewer #1: All comments have been addressed

Reviewer #2: All comments have been addressed

2. Is the manuscript technically sound, and do the data support the conclusions?

Reviewer #1: (No Response)

Reviewer #2: (No Response)

3. Has the statistical analysis been performed appropriately and rigorously? 

Reviewer #1: (No Response)

Reviewer #2: (No Response)

4. Have the authors made all data underlying the findings in their manuscript fully available?

Reviewer #1: (No Response)

Reviewer #2: (No Response)

5. Is the manuscript presented in an intelligible fashion and written in standard English?

Reviewer #1: (No Response)

Reviewer #2: (No Response)

6. Review Comments to the Author

Reviewer #1: (No Response)

Reviewer #2: (No Response)

7. PLOS authors have the option to publish the peer review history of their article (what does this mean?). If published, this will include your full peer review and any attached files.

Reviewer #1: No

Reviewer #2: No

---

## [Editor Report · Acceptance letter]

6 Oct 2022

PONE-D-22-02168R1 

Cardiovascular effects of exercice induced dynamic hyperinflation in COPD patients – dynamically hyperinflated and non-hyperinflated  subgroups 

Dear Dr. Varga:

I'm pleased to inform you that your manuscript has been deemed suitable for publication in PLOS ONE. Congratulations! Your manuscript is now with our production department. 

Kind regards, 

on behalf of

Dr. Nejka Potocnik 

Academic Editor

PLOS ONE